# *Sleeping Beauty* Transposon Insertions into Nucleolar DNA by an Engineered Transposase Localized in the Nucleolus

**DOI:** 10.3390/ijms241914978

**Published:** 2023-10-07

**Authors:** Adrian Kovač, Csaba Miskey, Zoltán Ivics

**Affiliations:** Transposition and Genome Engineering, Research Centre of the Division of Hematology, Gene and Cell Therapy, Paul Ehrlich Institute, Paul Ehrlich Str. 51–59, D-63225 Langen, Germany; adrian_kovac@yahoo.com (A.K.); csaba.miskey@pei.de (C.M.)

**Keywords:** genetic engineering, targeted integration, nucleolus, ribosomal RNA, transposition, chromosomal integration

## Abstract

Transposons are nature’s gene delivery vehicles that can be harnessed for experimental and therapeutic purposes. The *Sleeping Beauty* (SB) transposon shows efficient transposition and long-term transgene expression in human cells, and is currently under clinical development for gene therapy. SB transposition occurs into the human genome in a random manner, which carries a risk of potential genotoxic effects associated with transposon integration. Here, we evaluated an experimental strategy to manipulate SB’s target site distribution by preferentially compartmentalizing the SB transposase to the nucleolus, which contains repetitive ribosomal RNA (rRNA) genes. We generated a fusion protein composed of the nucleolar protein nucleophosmin (B23) and the SB100X transposase, which was found to retain almost full transposition activity as compared to unfused transposase and to be predominantly localized to nucleoli in transfected human cells. Analysis of transposon integration sites generated by B23-SB100X revealed a significant enrichment into the *p*-arms of chromosomes containing nucleolus organizing regions (NORs), with preferential integration into the p13 and p11.2 cytobands directly neighboring the NORs. This bias in the integration pattern was accompanied by an enrichment of insertions into nucleolus-associated chromatin domains (NADs) at the periphery of nucleolar DNA and into lamina-associated domains (LADs). Finally, sub-nuclear targeting of the transposase resulted in preferential integration into chromosomal domains associated with the Upstream Binding Transcription Factor (UBTF) that plays a critical role in the transcription of 47S rDNA gene repeats of the NORs by RNA Pol I. Future modifications of this technology may allow the development of methods for specific gene insertion for precision genetic engineering.

## 1. Introduction

Technologies of genome engineering, including those applicable for gene therapy, have made great advances over the last decade. The development of designer nucleases, especially the CRISPR/Cas9 system, has made it possible to precisely engineer genomes with high efficiency and specificity [1]. Those applications of designer nucleases that aim to introduce foreign DNA into a genomic site almost exclusively depend on the generation of a double-strand break (DSB) at the targeted site, and gene addition at that DSB is primarily the outcome of the homology-dependent repair (HDR) pathway of the target cells [2]. There are two major limitations of this reaction. First, designer nucleases often introduce DSBs at off-target sites (that is, sites that are not targeted by design), which is a safety issue in the context of therapeutic applications. Second, the efficiency of gene addition is limited when nucleases are used to generate knock-ins, because HDR is generally less efficient than non-homologous end-joining (NHEJ) [3], especially for large transgenes [4]. Homologous recombination is in fact entirely inactive in some cell types and cell cycle phases [5,6,7], thereby limiting HDR-dependent gene knock-ins in certain applications.

Some of the limitations of nuclease-based technologies do not apply to integrating vectors based on viruses or transposons, which have specifically evolved to integrate genetic cargo into target genomes. For example, integration by these genetic elements is executed by concerted transesterification reactions [8], which are not dependent on the generation of DSBs in the target genome. This avoids activating the DNA damage response of the target cells [9,10] and prevents the generation of unwanted byproducts of DSB repair [5]. One drawback of integrating vectors is, however, that they lack specificity: the integration of the genetic cargo occurs at a site in the target genome, which is selected in a biased, but unspecific manner [11]. This can make the effect of the delivered transgene hard to predict, as the chromatin environment can have an impact on the expression level or whether the transgene is expressed at all. Even more critically, several integrating vectors, particularly vectors of viral origin, have biased integration profiles that increase the chance of integration in or near expression units, thereby simultaneously increasing the chance of disrupting and/or deregulating coding or regulatory sequences in the target genome [12]. In the worst case, disruption of tumor suppressor genes or activation of proto-oncogenes can result in the transformation of the target cell and, in a therapeutic context, tumor formation in the patient [13,14].

*Sleeping Beauty* (SB) is a synthetic transposon that was reconstructed based on sequences of transpositionally inactive elements isolated from fish genomes [15]. SB is the most thoroughly studied vertebrate transposon to date, and it supports a full spectrum of genetic engineering applications, including the generation of transgenic cell lines, induced pluripotent stem cell (iPSC) reprogramming, phenotype-driven insertional mutagenesis screens in the area of cancer biology, germline gene transfer in experimental animals and somatic gene therapy both ex vivo and in vivo [reviewed in [16,17,18,19,20,21,22,23,24,25]]. SB preferentially inserts into TA dinucleotides with a close-to-random genome-wide profile [26,27,28,29,30], which may enable safer therapeutic cell engineering than with integrating viral vectors. However, even the use of vectors with close-to-random integration profiles carries a finite risk of insertional mutagenesis. Thus, it is of great interest to engineer integrating vectors in ways that reduce the risk of genotoxic effects.

One way of improving the safety profile of semi-randomly integrating gene delivery systems is to artificially alter their target site selection properties to make them preferentially integrate at predetermined sites into the genome or at least reduce the likelihood of integration into “dangerous” loci. Attempts at retargeting both viral and transposon-based integrating vectors have been made, with varying degrees of efficiency [reviewed in [12]]. For example, mutant versions of retroviral integrase (IN) were shown to target integrations away from transcriptional regulatory genes by disrupting a targeting mechanism based on the interaction of IN with BET proteins through mutagenesis of IN [31]. Transposase proteins have mainly been engineered by attaching foreign DNA binding domains (DBDs) to them in the expectation that transposon integrations would be biased towards sites specified by these DBDs [reviewed in [12]]. Designer DBDs with programmable DNA binding specificities, including zinc fingers (ZFs) [32], transcription activator-like effectors (TALEs) [33] and the CRISPR/Cas system [1] have been explored to predefine target sites of transposons; notably, the SB and the *piggyBac* (PB) systems have been successfully targeted to a range of exogenous or endogenous loci in the human genome [[34,35,36,37,38,39,40,41] and reviewed in [12]]. However, a consistent finding across all targeted transposition studies is that while some bias can be introduced to the vector’s integration profile, the number of targeted integrations is relatively low when compared to the number of untargeted background integrations [12].

An interesting type of target site has so far been mainly absent from retargeting studies: highly expressed multicopy genes like tRNA and rRNA genes. Targeting this type of gene would have several advantages. Due to the redundant nature of the gene, disruption of any one copy of it would not cause pronounced negative effects [42], while the active transcription at these loci would prevent transgene silencing, thereby enabling the expression of foreign gene constructs post-integration [43]. In fact, multicopy housekeeping genes are targeted by several naturally occurring transposable elements [12] and targeting of tRNA sites has independently arisen several times over the course of natural evolution [44], for example by the Ty1 and Ty3 elements.

Targeting of rRNA genes has also evolved naturally, for example by the R1, R2 and *Pokey* elements [45,46,47]. The rRNA genes are found as tandem repeats on the short arms of the acrocentric chromosomes 13, 14, 15, 21 and 22 [48] (Figure 1A). The repeated sequences situated within these short arms are also known as nucleolus organizer regions (NORs) [49,50,51], because the nucleoli, the subnuclear compartments of ribosome assembly, form around them [52]. Each repeat unit of rDNA contains RNA Pol I-driven genes encoding the 18S, 5.8S and 28S rRNAs (while the 5S rRNA is transcribed from a different locus) (Figure 1B) [53,54], collectively called the ribosomal DNA (rDNA), which is transcribed in the nucleoli that form at the end of mitosis [55]. Each diploid human cell has about 600 copies of the rRNA genes [42] flanked by additional sequences called the proximal and distal junction (PJ and DJ), making up the NORs (Figure 1A) on the five chromosomes mentioned above. Owing to the wealth of rRNA genes and the presence of spacers between the gene repeats that likely confine natural insulator functions [56], the rDNA is an appealing safe harbor for transgene integration.

The above-mentioned transposons that are naturally targeted to rDNA all integrate into the 28S rRNA gene (Figure 1B). An attempt to artificially target recombinant adeno-associated viral (rAAV) vector integrations to rDNA has been made by including regions homologous to rDNA sequences in the transgene cassette in order to induce homologous recombination at rDNA loci [57,58]. This effect was hypothesized to be supported by the fact that AAV genomes are normally processed at nucleoli [59] and that rAAV vectors have a higher-than-random chance to be integrated into rDNA [60,61]. Both Wang et al. [57] and Lisowski et al. [58] have established an increased frequency of rAAV integration into rDNA after in vivo delivery into experimental mice. However, because rAAV vectors are not equipped to promote genomic integration, the overall frequencies of locus-specific vector integration remained low in these studies. In addition, it was shown that a fusion protein composed of the Human Immunodeficiency Virus 1 (HIV-1) IN and the homing endonuclease I-PpoI, which has a naturally occurring recognition site in the 28S rRNA genes in eukaryotes, targeted lentiviral vector integration into rDNA with an efficiency of ~3% [62]. Finally, a recent report showed enhanced integration into rDNA by an engineered PB transposase containing a nucleolar localization signal (NoLS) in Chinese Hamster Ovary (CHO) cells [63].

Here, we attempted to generate an engineered SB transposon vector system that preferentially integrates into rDNA or NOR regions by physically localizing the transposase to nucleoli and demonstrate biased integration into nucleolar domains of human chromosomes enabled by a nucleophosmin (B23)-SB transposase fusion protein. This work serves as a proof of concept for this novel retargeting strategy and demonstrates that it merits further investigation and refinement.

## 2. Results

### 2.1. A B23-SB100X Fusion Protein Retains Transpositional Activity and Is Localized to the Nucleolus

In order to increase the frequency of insertions into rDNA or NOR regions, we aimed to generate an SB transposase variant that preferentially localizes to the nucleolus. This was first attempted by fusing several short NoLSs to the N-terminus of the hyperactive transposase SB100X. These included HIV Tat (MGRKKRRQRRRAHQ) [64], HIV Rev (MGRQARRNRRRRWRERQRQ) [65], human p120 nucleolar protein (MGSKRLSSRARKRAAKRRLG) [66] and HTLV Rex (MGPKTRRRPRRSQRKRPPTP) [67]. However, all the tested fusion proteins lost most of their transpositional activity and were not found to localize to the nucleolus (Appendix A). This led us to hypothesize that the N-terminus of SB100X might represent a poor environment for short localization sequences and decided to use a full-length nucleolar protein to achieve nucleolar localization.

We next attempted to employ a full-length nucleolar protein, and we selected B23 (also known as nucleophosmin), which is involved in ribosome biogenesis, associated with ribonucleoprotein structures [68] and localized to the granular component of the nucleolus [69]. We fused the entire sequence of the B23 protein to the N-terminus of SB100X (C-terminal additions to SB transposases were previously shown to completely abolish transpositional activity [34,36,70]), where the two fusion partners were separated by a 14 amino acid linker of the sequence KLGGGAPAVGGGPK, which has previously been used to fuse various protein domains to the N-terminus of SB transposase and shown to enable maintenance of transpositional activity [35,36,40]. The amino acid sequence of the fusion protein is shown in Appendix A. Expression of the fusion protein was tested using a Western blot following transient transfection into human HeLa cells (Figure 2A), which showed that the hybrid transposase is expressed and has the expected size of ~77.2 kDa.

After confirming the proper construction and expression of the fusion transposase, its transpositional activity was tested in a colony formation assay in HeLa cells in combination with an antibiotic resistance gene-tagged transposon. This assay revealed high transpositional activity of the fusion transposase, despite the relatively large size of B23 (29.5 kDa) (Figure 2B and Appendix A).

In order to work as intended, the fusion protein needs to localize to the nucleoli when expressed in cells. This was tested by immunofluorescence microscopy, using fluorescence-tagged B23 as a marker for nucleoli (Figure 2C). As expected, and consistent with the presence of a nuclear localization signal (NLS) in the SB transposase [71], unfused SB100X was distributed across the entire nucleus, while it was depleted in nucleoli (Figure 2C). The B23-SB100X fusion displayed a distinct subnuclear localization; while also being spread across the entire nucleus, it was enriched in nucleoli, rather than being depleted there (Figure 2C). However, some differences between the distribution of B23-SB100X and unfused B23 remained; while B23 was found in nucleoli of all cells (Figure 2C), the B23-SB100X fusion localized to the nucleolus in only a subset of cells, and in others it was distributed similarly to unfused SB100X. Additionally, as mentioned above, B23-SB100X was enriched in, but not exclusive to, the nucleolus, while B23 was exclusively detected in the nucleolus.

### 2.2. Sleeping Beauty Transposon Integrations Catalyzed by B23-SB100X Preferentially Occur within and in the Vicinity of Nucleolar Chromatin

After confirming nucleolar localization of B23-SB100X, the genomic distribution of insertions catalyzed by this fusion transposase was compared to the insertion pattern obtained with unfused SB100X. This comparison was made for two different transposon constructs containing a neomycin selection marker (Appendix A). In one construct, the neomycin gene was driven by the SV40 (RNA Pol II) promoter (pT2/SV40-neo), while in the other construct, the SV40 promoter was replaced with a 45S RNA (RNA Pol I) promoter and an internal ribosomal entry site (IRES) (pT2/HENA-neo) [72]. We considered the use of an RNA Pol I promoter potentially beneficial because of the high abundance of RNA Pol I in the nucleolus. Additionally, we hypothesized that the interaction between RNA Pol I molecules in the nucleolus and the 45S promoter might result in enhanced targeting due to a tethering effect.

To determine the insertion sites of the transposons, we harnessed a well-established, targeted library preparation method to selectively amplify the transposon-genomic DNA junctions form the genomes of the transfected HeLa cells [see, for example, [40]]. Briefly, the procedure involves random shearing of the genomic DNA by sonication, ligation of linkers to the ends of the broken DNA, PCR amplification by employing transposon- and linker-specific primers and high throughput sequencing of the products. Recently, the complete human genome was published, including the *p*-arms of acrocentric chromosomes, which had for a long time belonged to the “dark matter” of the previous genome assemblies [73]. However, unambiguous mapping of these regions is still challenging due to the repetitive nature of these sequences. We made use of long (2 × 150 bp) Illumina sequencing to address this difficulty. After merging the paired-end reads to maximize the read length, the mapping identified a large number of unique insertion sites, which were supported by at least five independent reads: SB100X + pT2/SV40-neo, 181,796; B23-SB100X + pT2/SV40-neo, 119,087; SB100X + pT2/HENA-neo, 247,255 and B23-SB100X + pT2/HENA-neo, 289,688 insertions. As a quality control, transposon integration sites were visualized by SeqLogo analysis that revealed the expected, preferred integration sites of SB consisting of the 8-bp palindromic AT repeat sequence ATATATAT centered on the actual TA target dinucleotide (underlined) (Appendix A).

We next investigated the impact of the B23-transposase fusion on the distribution of the insertion sites of the transposons, in which the antibiotic marker gene was driven either by the HENA or by the late SV40 promoters, using transposition assays with the unfused SB100X transposase as a control. We detected a 64% (HENA) and a 52% (SV40) increase in the number of insertions (*p* < 0.0001, Fisher’s exact test) in the *p*-arms of the NOR-containing chromosomes, if the transposition reactions were initiated by the B23-SB100X fusion protein (Figure 3A). Contrarily, both transposase versions inserted the transposons into the *p*-arms of other chromosomes, which are not constituents of the nucleolus, at the same frequency (Figure 3A). These results suggest that the insertions of both transposon versions could efficiently be targeted to nucleolar DNA if the transposase is directed to the nucleolus by the B23 tag.

The *p*-arms of the acrocentric chromosomes contain three distinct cytobands identified by Giemsa-staining: p13, p12 and p11.2. Among these, the central p12 segments include the NORs (Figure 3B). When counting the insertions in these regions of all the acrocentric chromosomes, we found an up to 107% increase (p13, B23-SB100X, pT2/SV40neo) in the number of insertions in the cytobands surrounding the NOR region in the presence of the fusion transposase (*p* < 0.001, Fisher’s exact test) (Figure 3B). Intriguingly, no significant difference was detected in the insertion numbers between the conditions within the NOR (p12) segments of the chromosomes (Figure 3B). Analysis of the insertion frequencies on each *p*-arm of the acrocentric chromosomes showed that the most efficient targeting was achieved on chromosome 15, where a 144% increase was detected in the number of insertions in the presence of the B23-SB100X transposase fusion (*p* = 9.48 × 10^−6^, Appendix A). Studying the distribution of the transposon insertions within the acrocentric *p*-arms revealed no apparent hotspots for integrations. We found that the NORs (marked by the p12 cytobands) are TA-poor (thus, per se transpositionally disfavored for SB that strongly requires TA dinucleotides for insertions [74]) segments of the acrocentric *p*-arms, compared to the neighboring cytobands. Nevertheless, the frequency of integrations in p13 and p11.2 bands did not show a correlation with their TA contents (Figure 3C).

Next, we investigated insertion frequencies in genome segments that are in physical proximity to the nucleolar DNA. The nucleolus tends to associate with predominantly heterochromatic, late-replicating, often sub-telomeric genome segments (known as nucleolus-associated chromatin domains, NADs), which are poor in genes and rich in transposable elements [75]. We assumed that these regions could also be targeted by the relocalized transposase due to the adjacency of NADs to the peripheral nucleolar DNA. Therefore, we quantified the insertions in NADs identified in HeLa cells and in human embryonic fibroblasts. We found that transposons harboring either of the promoters integrated significantly more frequently (*p* < 0.0001 Fisher’s exact test) into NADs if the transposase was fused to B23, compared to the insertions generated by unfused SB100X transposase (Figure 4A).

The nucleolus is also involved in organizing heterochromatin by aggregating with chromatin close to the nuclear lamina, known as lamina-associated domains (LADs). NADs were found to be strikingly associated with LADs. The chromatin segments in association with lamin A/C and B1/B2 (LMNA and LMNB) are largely overlapping, but in HeLa cells, it was found that about 30% of the LADs are uniquely associated with LMNB [76]. Thus, we studied how these NAD-neighboring chromatin domains could be targeted by transposon insertions. The analyses showed that B23-SB100X significantly increased the insertion abundance by up to 30%, in both LMNA and in LMNB, as compared to insertions catalyzed by SB100X (*p* < 0.0001, Fisher’s exact test, Figure 4B). Collectively, these results implied that the targeted chromatin domains for integrations were not restrained to the nucleolar DNA, but the increased transposition frequency also applied to a nuclear territory near the nucleolus.

Next, we studied the distribution of insertion sites neighboring and within the rRNA genes. The Upstream Binding Transcription Factor (UBTF) protein plays a critical role in transcription of the 47S rDNA gene repeats of the NORs by RNA Pol I. We analyzed the insertion frequencies in regions of UBTF-bound DNA, suggested by significant peaks using raw data of previous ChIP-Seq experiments. We measured integration frequencies in the ChIP-Seq peaks of Subunit A of RNA Pol II and III (POLR2A and POLR3A), respectively, as controls. Figure 5A shows that both transposon constructs displayed an enrichment in integration into the regulatory regions of RNA Pol I, marked by UBTF, in the presence of the B23 tag. The increase in insertion numbers was larger (46%) if the transposons carried the SV40 promoter (*p* < 0.001, Fisher’s exact test). No significant change in integration frequencies was detected in RNA Pol II and III promoter regions (Figure 5A).

Finally, we quantified the insertions in the gene bodies of the 47S rRNA genes (large and small subunit rRNA genes located in the nucleolar DNA) and in the 5S rRNA genes localized elsewhere in the genome. We could identify only a low number of insertions, most likely due to the difficulty of unambiguous mapping within these repetitive regions of low sequence variability. Nevertheless, we found significantly more integrations of the HENA-promoter-containing transposons if these were co-transfected with the B23-SB100X transposase (*p* < 0.01) (Figure 5B). No significant differences were detected if the marker genes in the transposons were driven by the SV40 promoter and/or in the RNA Pol II transcribed (non-nucleolar) 5S rRNA genes (Figure 5B). These findings may imply that transgene expression from gene bodies of RNA Pol I transcription units could be improved by harnessing an RNA Pol I promoter driving transgene expression. Studying the relative distributions of the insertions between strictly nucleolar DNA (represented by the *p*-arms of the acrocentric chromosomes) and the chromatin near to the nucleolus revealed that integrations into NAD and LAD segments were mainly responsible for the targeting effect (Appendix A). Taken together, the findings above indicate that the relocalization of the transposase protein to the nucleoli induced a redirection of transposon insertions to nucleolus-related DNA.

## 3. Discussion

Targeting foreign gene constructs into genomic safe harbors (GSHs) fulfills two equally important requirements in the context of human applications: efficient expression of the transgene so that a long-term therapeutic benefit can be elicited in the genetically engineered cell population and integration into a region of the genome where the endogenous transcriptional program of the cell is not disturbed. GSHs can be bioinformatically allocated to chromosomal sites or regions if they satisfy the following criteria: (i) no overlap with transcription units, (ii) a distance of at least 50 kb from the 5′-end of any gene, (iii) at least 300 kb distance to cancer-related genes and (iv) microRNA genes, and (v) regions outside of ultra-conserved elements (UCEs) [77,78]. Paradoxically, the GSH sites that are traditionally targeted in most applications, including AAVS1, CCR5 and ROSA26, do not actually fulfill all GSH criteria, because these sites by definition map into transcription units. Conversely, the safest GSH sites far away from genes might not support high levels of expression [79]. As a target for transgene integration, rDNA seems like an interesting, non-canonical GSH candidate, owing to the many unique features it bears in comparison with non-nucleolar DNA. First, rRNA genes are isolated on five short chromosome arms where they reside far away from protein-coding genes with oncogenic potential. Second, a loss-of-function mutation due to transgene insertion into any one rRNA gene can be compensated for by the repetitive nature of rRNA genes. Third, the spacer regions between rRNA gene repeats may act as chromatin insulators and limit the spreading of chromatin between the transgene and the surrounding chromosomal regions.

Apparently, certain mobile genetic elements evolved a natural ability to insert into rRNA genes, likely driven by selection for mechanisms that allow expression (and hence propagation) of the transposon with limited negative impact on the host cell. The best characterized examples are the R1 and R2 retrotransposons from the silkworm *Bombyx mori* (R1Bm and R2Bm) that insert themselves into the 28S rRNA genes in a highly sequence-specific manner. These retrotransposons express an endonuclease that can specifically cleave the 28S rRNA gene and the resulting nick can be employed by these retrotransposons for integration at that site [80,81]. The involvement of a sequence-specific endonuclease in the transposition reaction of certain retroelements suggests that these elements could possibly be redirected to distinct sequences by engineering the domain of the endonuclease that is responsible for DNA recognition. However, unlike non-LTR retrotransposons, SB and other DNA transposons do not rely on DNA cleavage for their integration; instead, genomic integration is the result of direct attacks of the two target DNA strands by 3′-OH groups exposed at the transposon ends without the involvement of prior cleavage [8]. This, together with the fact that the only strict requirement for SB integration is a TA dinucleotide, indicates that the engineering of an SB transposase to artificially evolve specificity for a given DNA sequence would be extremely difficult. Instead, foreign DBDs have been fused to the SB transposase in the hope that at least a fraction of transposition events would be biased towards sites specified by these DBDs [reviewed in [12]]. Previous work established that although such bias can be accomplished, the efficiency of targeted integration remained low [reviewed in [12]].

Here, we tested a thus far unexplored concept for targeting SB transposon integrations into select genomic regions. Instead of fusing protein domains to the transposase that specify binding to a particular DNA sequence in the genome, we present here a proof of concept for a strategy of directing transposon insertions to specific chromosomal domains by influencing the subnuclear localization of the transposase. Several measures indicate that the fusion of SB100X to the nucleolar protein B23 results in an altered insertion profile that favors nucleolar DNA and adjacent regions. On the largest scale, the B23-SB100X fusion transposase favored insertion to the *p*-arms of rDNA-containing acrocentric chromosomes when compared to unfused SB100X (Figure 3A). This finding, combined with the lack of enrichment into the *p*-arms of non-acrocentric chromosomes (those not harboring rDNA) suggests that the localization of the transposase in nucleoli was responsible for this altered insertion pattern. It is also striking that enrichment occurred for the *p*-arms of all five acrocentric chromosomes and for both transposons tested, suggesting a robust targeting of these regions. Analysis on a smaller scale indicated that enrichment did not, as expected, occur mainly in the NORs themselves, as evidenced by the relative enrichment of insertion into cytobands p11.2 and p13, but not into p12 (Figure 3B,C). Instead, most of the observed enrichment occurred in chromosomal regions adjacent to the NORs, while the NORs themselves were generally disfavored integration sites and little enrichment could be observed here.

Further support for the hypothesis that insertions with B23-SB100X will often occur in genomic regions that are localized close to the nucleolus, even if it might not occur in the NORs themselves, is provided by the enrichment in NADs and LADs (Figure 4). Both of these nuclear domains are closely associated with the nucleolus and might therefore be targeted by B23-SB100X as a bystander effect, especially if SB integration into NORs is disfavored.

While analyzing the NORs—represented by cytoband p12—as a whole, although no enrichment could be observed, some other measures suggest that a minor targeting effect into rDNA might be present. When analyzing the relative insertion frequencies into regions associated with RNA Pol I, which transcribes the rDNA genes, it can be shown that the region is targeted more frequently by B23-SB100X than unfused SB100X, while the same effect cannot be observed for RNA Pol II and III (Figure 5A). A direct comparison of reads mapping to the 47S pre-RNA gene itself also showed an enrichment with B23-SB100X, although this can only be observed for the transposon containing the RNA Pol I promoter (Figure 5B). Taken together, these observations suggest that while the overall (detectable) enrichment into NORs is poor, some subregions might be targeted. We consider two main explanations for the low enrichment—and overall low insertion frequency—into NORs. On the one hand, it is possible that the lack of detectable enrichment in NORs is caused by the overall difficulty of detecting and precisely mapping insertions into these regions compared to the less redundant adjacent regions. It is possible that many targeted insertions occur in NORs but it is difficult to detect them, while the non-NOR regions of the *p*-arms, which would also be localized close to nucleoli, still receive significant enrichment and allow easier detection of the targeting effect. Another possible reason for the lack of detectable enrichment and overall low insertion frequency into NORs is the comparably low TA content of these regions. Compared to adjacent genomic regions, the NORs have a low TA content (Figure 3C), which makes them poor targets for SB insertion. This could indicate that, although B23-SB100X is localized to the nucleolus, the bulk of catalyzed insertions do not occur into the NORs themselves due to their low abundance of TA sites. Instead, the fusion transposase would preferentially integrate in regions close to—but not part of—the NORs. It is also possible that the observed integration pattern results from a combination of both described effects—a low frequency of insertion into NORs due to a low TA content and difficult mapping of those insertions due to the repetitive nature of the NORs. However, even if the true enrichment of insertions into NORs is relatively weak due to the limited availability of target sites, and most enrichment occurs into NOR-adjacent regions, it could still provide a safety benefit. Indeed, while the NORs, due to the redundant nature of rDNA, present particularly attractive targets for insertion, the *p*-arms of the acrocentric chromosomes are overall gene-poor when compared to the *q*-arms [73], therefore targeting of the *p*-arms themselves would by itself reduce the risk of genotoxicity.

One finding consistent across most analyses in this study is that the observed enrichment occurred for both transposons (containing either RNA Pol I or Pol II promoters), and seems to be exclusively tied to the presence of B23 in the fusion transposase. This refutes the hypothesis that the use of an RNA Pol I promoter would improve targeting due to tethering to RNA Pol I molecules. Additionally, this result suggests that expression by RNA Pol II of genes situated in these genomic regions is possible or that the limiting factor of detecting insertions in NORs is not related to transgene expression.

Another aspect that might influence expression of genes inserted into or near NORs is the phenomenon of nucleolar dominance (ND), which is observed across a broad range of organisms, including human cells [82,83]. ND is an epigenetic phenomenon in which entire NORs are silenced as a means of rRNA dosage control [84]. As the mechanism of ND is not yet completely understood, and seems to be highly divergent between species [85], it might be assumed that some genes integrated into or near NORs by the transposase could be silenced in a bystander effect, as ND seems to act in a locus-wide manner [85]. However, it has been suggested that in HeLa cells, NORs on chromosomes 15 and 22 are more likely to be silenced than those from chromosomes 13, 14 and 21 [82], and neither enrichment nor overall insertion numbers seemed to be particularly reduced for these chromosomes in our experiments, indicating that ND did not have a significant negative impact on the targeting of NORs. We note that under the experimental conditions used in our studies, we expect multiple transposon integrations, on the order of 2–6 per transgenic cell [86]; thus, potential, locus-specific silencing of any given integration can be masked by other, expressed transgenes residing in the same cell.

As the overall targeting efficiency shown here is significantly lower than what would be necessary for any relevant applicability, it is clear that modifications need to be made to address this limitation. It seems likely that the incomplete relocalization of B23-SB100X is one of the main reasons for the low rate of insertion into NORs. While unfused B23 is exclusively localized to the nucleoli, B23-SB100X is enriched in nucleoli, but also found spread throughout the nucleoplasm (Figure 2C). This means that a large number of transposon insertions can occur into non-nucleolar DNA, resulting in the observed high background. Nucleolar localization might be improved by replacing B23 with a different localization domain and/or by changing the overall architecture of the fusion protein, e.g., by using alternative interdomain linkers. Apart from improving localization to nucleoli, an attractive strategy might consist of combining the sub-nuclear localization-based retargeting explored here with the DNA sequence-based approaches described previously. It is possible to envision a combined system in which the transposase contains both a nucleolar localization signal and a DBD specifically binding to an rDNA sequence. The nucleolar localization signal would mediate transport of the transposase to the nucleolus, where the DBD would mediate sequence-specific insertion into rDNA. This might serve to synergistically combine sequence- and localization-based targeting approaches into an entirely novel targeting mechanism. Due to their inherent flexibility, RNA-guided targeting mechanisms [40] might be particularly useful for this approach.

## 4. Materials and Methods

### 4.1. Generation of the B23-SB100X Fusion

An expression plasmid for B23-SB100X was generated by inserting the B23 coding sequence and a linker at the N-terminus of the SB100X coding sequence in a vector where SB100X expression is driven by a CAGGS promoter. The backbone was generated by amplifying the entire vector with the primers L-SB100X_fwd1 and NotI-SBT7_rev, then amplifying the resulting product with FseI-L-SB100X_fwd2 and NotI-SBT7_rev. This removes the start codon from SB100X, introduces a linker with the sequence KLGGGAPAVGGGPK at the N-terminus of SB100X and introduces an *Fse*I site at the 5′-end and a *Not*I site at the 3′-end, respectively. The insert was generated by amplifying the B23 sequence from the plasmid pDsRed-B23 (addgene #34553) using the primers NotI-S-B23_fwd and FseI-B23_rev, which introduce a start codon at the 5′-end of the B23 sequence and *Not*I and *Fse*I sites at the 5′-end and the 3′-end, respectively. The B23 sequence was inserted into the linearized plasmid by ligation with T4 DNA ligase. All primer sequences are provided in Appendix A.

### 4.2. Western Blot

For Western blots, 10^6^ HeLa cells were transfected with 2 µg of B23-SB100X or SB100X expression plasmids using polyethyleneimine. Cells were harvested 48 h after transfection, lysed with RIPA buffer, DNA was sheared and cellular debris was removed by centrifugation. A total of 30 µL of each extract was loaded onto denaturing SDS gels (10% separating gel) and separated at 200V for 1 h. Proteins were transferred on a nitrocellulose membrane and developed with α-SB (RRID: AB_622119, 1:500) for 1.5 h and with α-goat-HRP (RRID: AB_258425, 1:1000) for 1 h.

### 4.3. Colony Formation Assay

All transposon vectors were based on the pT2 plasmid (addgene #26557). To test for transposition activity, 10^6^ HeLa cells were transfected with 50 ng of either SB100X or B23-SB100X expression plasmids and 500 ng of the transposon plasmid pT2/puro, using polyethyleneimine. Cells were seeded onto 10 cm dishes and cultured in DMEM including 10% FCS, 2 mM L-Glutamine and penicillin/streptomycin and selected with puromycin at 1 µg/mL. Colonies were allowed to grow for 11 days, then fixed with 4% PFA and stained with methylene blue. Cells were counted using the colony counter plugin of ImageJ v1.52 (National Institutes of Health, Bethesda, MD, USA) using the following settings: minimal size 150 px and circularity 0.85–1.

### 4.4. Immunofluorescence Microscopy

For localization analysis, 10^6^ HeLa cells seeded on coverslips were transfected with 1 µg of pDsRed-B23 (as a nucleolar marker fused to a red fluorescent protein, addgene #34553) and 1 µg of B23-SB100X or 1 µg of SB100X expression plasmids. At 72 h after transfection, cells were fixed for 30 min with 4% PFA in PBS, washed with PBS, incubated and washed with 100 mM glycine in PBS, permeabilized with 1% BSA + 0.1% Triton X-100 in PBS for 30 min and blocked for 1 h in 3% BSA in PBS. Samples were then incubated for 1 h with α-SB (RRID: AB_622119, 1:100 in 1% BSA), washed with PBS-T and incubated with α-goat-Alexa488 (RRID: AB_228313, 1:1000) and DAPI (1:10,000) in 1% BSA in PBS for 1 h in darkness. Samples were then mounted on microscopy slides and analyzed using a T*i* Eclipse Inverted Microscope (Nikon, Tokyo, Japan).

### 4.5. Generation of Integration Libraries

SB insertions were generating by transfecting ~1.3 × 10^6^ HeLa cells with 300 ng of SB100X or 900 ng of B23-SB100X in combination with 6 µg of the transposons pT2/HENA-neo or pT2/SV40-neo. Two days after transfection, selection was started by culturing cells in DMEM + 10% FCS supplemented with 1mg/mL neomycin. Cells were cultured in the presence of antibiotics for 16 days, then cells were harvested and gDNA (genomic DNA) was isolated using a column-based kit. Isolated gDNA was digested for several hours with *Dpn*I, then purified by gel electrophoresis and isolated using a Zymo Large Fragment Recovery Kit. Approximately 1 µg of purified DNA was sonicated to an average length of 600 bp using a Covaris M220 Ultrasonicator (Covaris, Woburn, MA, USA)at the following settings: Peak Power 50.0, Duty Factor 5.0, Cycles/Burst 200 and Duration 80 s. Sonicated DNA was then isolated using magnetic beads. End repair was performed using NEB End Repair Enzyme Mix, followed by dA-tailing using Klenow exo- in dA-tailing buffer (NEB). Linkers were generated by annealing the oligos Linker_TruSeq_T+ and Linker_TruSeq_T-, and linker ligation was performed using Blunt/TA Master Mix (NEB). Genome-transposon-junctions were amplified with a nested PCR using the following primer pairs: T-bal_long and TS_linker, then PE_nest_BC and SB20hmr_BC. All of the oligonucleotide sequences are provided in Appendix A. The thermocycler program used for the first PCR was 98 °C for 30 s; 10 cycles of 98 °C for 10 s, 72 °C for 40 s; 20 cycles of 98 °C for 10 s, 62 °C for 30 s, 72 °C for 30 s; and 72 °C for 5 min. The thermocycler program used for the second PCR was 98 °C for 30 s, then 20 cycles of 64 °C for 30 s, 72 °C for 30 s, then 72 °C for 5 min. Products of the second PCR were run on an ultrapure agarose gel and fragments with sizes between 200 and 500 bp were excised and purified from the gel. The libraries were sequenced on an Illumina NextSeq 550 instrument (Illumina, San Diego, CA, USA) with a 2 × 150 setting.

### 4.6. Integration Site Sequencing and Analysis

We used *seqkit* [87] for the initial read analysis as follows. The reads were demultiplexed and filtered for the presence of the IR-specific primer (allowing 6 mismatches). Filtering the reads for the presence of the IR-specific sequences downstream of the primer was used to ensure PCR-specificity. Reads containing plasmid backbone sequences following the IR were discarded. We used *fastp* 0.23.2 [88] to remove PCR duplicates from the reads and for subsequent quality-, adapter-, and minimum length- (≥30 bp) trimming. The same tool was used to merge the read pairs of the originally 2 × 150 bp reads, if possible. The first reads of the non-mergeable pairs were also kept for mapping the reads to the complete human assembly [T2T CHM 13v2.0/hs1, [73]] using *STAR* 2.7.10b [89] following the recommendation for ‘unique settings’ described previously [90]. The supporting reads for the mapped insertion sites were counted using *BEDTools* v2.30.0 [91]. Any locus supported by at least five independent reads was considered a valid insertion site. Integration sites are provided in Appendix A.

The analyses of the representation of insertion sites in various nucleolus-related genomic regions were performed in the *R* environment (https://www.R-project.org, R version 4.3.0, accessed on 1 April 2023) using the *genomation* package [92]. The position wait matrixes of the nucleotides around the insertion sites were plotted with the *SeqLogo* package. The acrocentric chromosome arms, the cytoband positions, and the coordinates of the ribosomal RNA genes (annotated in the repeat masker table) were downloaded from the UCSC Genome Browser for the *hs1* assembly (https://genome.ucsc.edu, T2T CHM 13v2.0, accessed on 1 April 2023).

The coverage on insertions on the acrocentric chromosomal arms was determined as follows. In order to address the difficulty of unambiguous mapping to repetitive regions (*p*-arms of the nucleolus forming chromosomes), we relaxed the criteria for valid insertion sites. We used ‘random’ settings [90] for *STAR*, which assigns the mapping locus of a read by picking one of the multiple possible (good quality) mapping positions, randomly. In addition, a single supporting read was sufficient to substantiate a mapping position as an insertion site. The segments defined by the cytobands coordinates for all nucleolar chromosomes were binned into 300 windows each, using *BEDtools*. After counting the insertions sites per bin, the insertion numbers were normalized by the RPKM (reads per kilobase per million) principle.

The coordinates of the nucleolus-associated domains (NADs) for human embryonic fibroblast and the HeLa cells genomes have been published [93,94]. The genomic positions of lamina-associated domains in HeLa cells have been described [95]. The coordinates of ChIP-Seq peaks of the transcription factors in HeLa cells can be retrieved from the Encode data collection (https://www.encodeproject.org, Stanford University, accessed on 1 April 2023) with the following accession numbers: ENCFF002CTD (POLIII) and ENCFF245CAL (POLR2A). Genomic coordinates of older assemblies were converted to hs1 using the *LiftOver* tool of the UCSC Genome Browser. To determine the binding sites of the UBTF transcription factor in the complete human genome, we downloaded the fastq files, which correspond to the UBTF ChIP-Seq reads and their controls, respectively, of the Encode projects ENCSR634ZGP and ENCSR760PBD. After mapping to the *hs1* with *STAR*, the narrow peaks were called with MACS2 1.4.2.20120305 [96] with standard settings. The coordinates of the significant peaks were expended by 500 nucleotides and used for the downstream analysis. We used the set of insertion sites determined by the ‘random’ settings (see above) to address the problem of unambiguous mapping to repeat regions (UBTF binding sites) and used these insertion site sets for quantification in all transcription factor-binding regions (ChIP-Seq peaks). We applied Fisher’s exact test to calculate whether the probability that a distribution of insertion sites within and outside of a region of interest is different from the hypothesized distribution arose by chance.

## Figures and Tables

**Figure 1 ijms-24-14978-f001:**
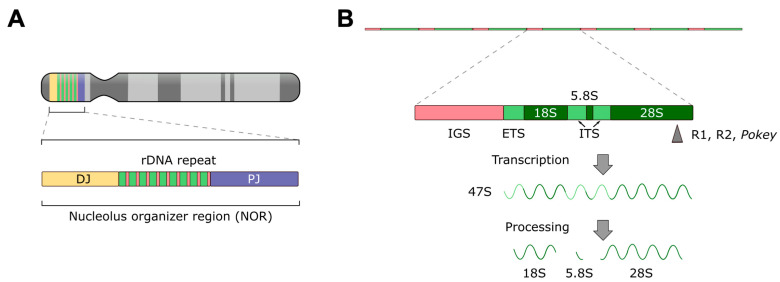
**Structure of eukaryotic rDNA and rRNA processing.** (**A**) Schematic representation of the organization of the NORs on the short arms of the acrocentric chromosomes. Proximal junction (PJ) and distal junction (DJ) regions make up the non-rDNA component of the NORs. (**B**) Schematic structure of rDNA loci. Repeats are tandemly arrayed, with the transcription units separated by intergenic spacers (IGS). The 18S, 5.8S and 28S rRNAs are separated by processing, which removes the external and internal transcribed spacers (ETS, ITS) from the transcript. Approximate insertion locations of R1, R2 and *Pokey* elements are indicated by a gray arrow.

**Figure 2 ijms-24-14978-f002:**
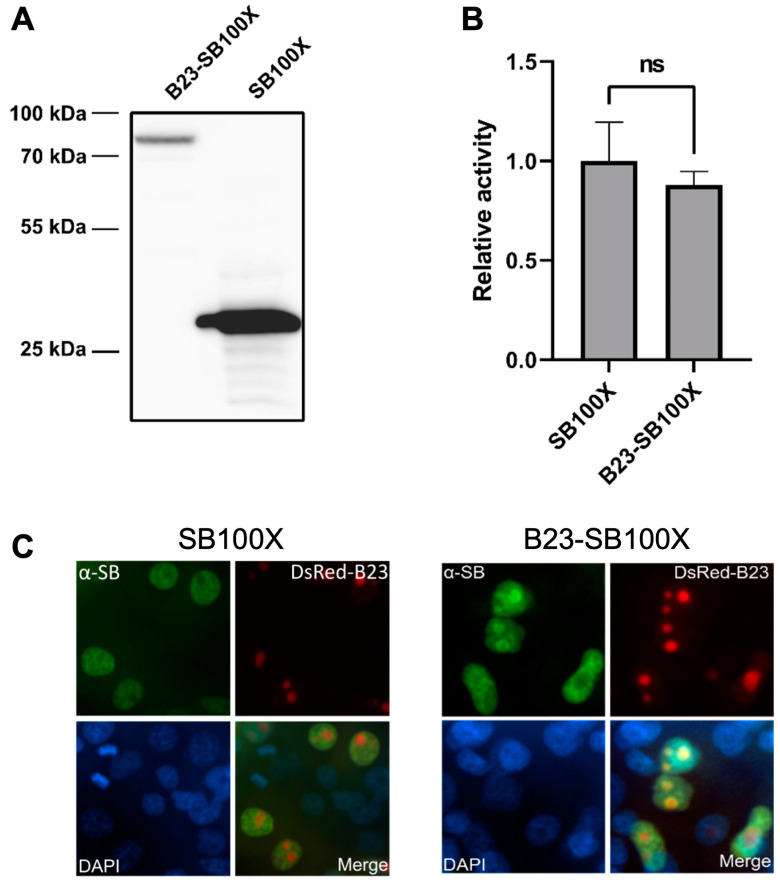
**Characterization of the B23-SB100X fusion transposase.** (**A**) Western blot of B23-SB100X and unfused SB100X with an α-SB antibody. Calculated sizes: SB100X, 39.3 kDa and B23-SB100X, 77.2 kDa. (**B**) Transposition activity of B23-SB100X relative to SB100X. Statistical significance tested with Welch’s *t* test, n = 3, ns = not significant. (**C**) Subnuclear localization of SB100X (left, green) and B23-SB100X (right, green) with DsRed-B23 as marker for nucleoli (red). Note that in some cells, B23-SB100X is enriched in the nucleolus (top left cells), while in others, it is depleted (bottom right cells), with the second distribution mirroring the usual localization of SB100X.

**Figure 3 ijms-24-14978-f003:**
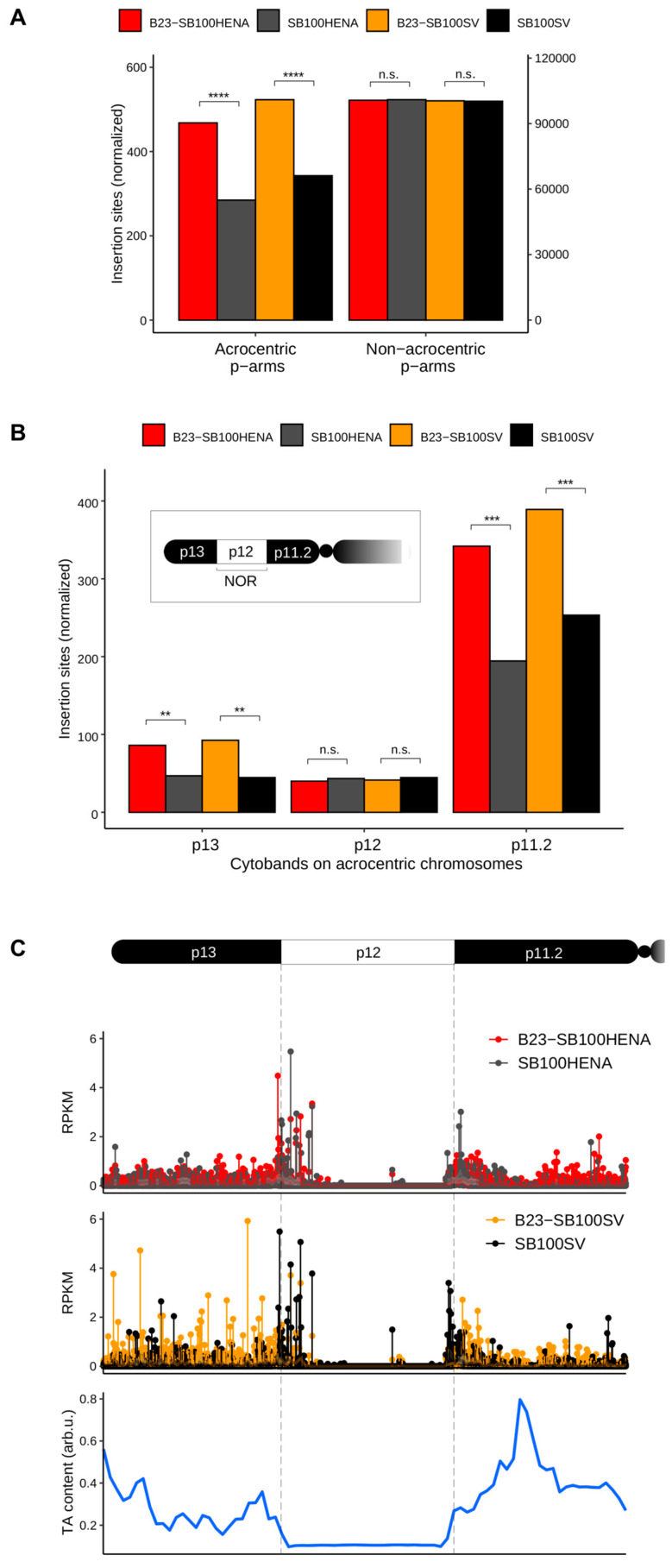
**Distribution of insertions in the *p*-arms of the chromosomes constituting the nucleolus.** (**A**) Cumulative numbers of insertions on the *p*-arms of the acrocentric chromosomes (13, 14, 15, 21 and 22) and on all the other chromosomes with *p*-arms. Insertion numbers are normalized with the total number of insertions per condition, shown on both axes. The second *y*-axis applies to the insertion sites on the non-acrocentric chromosomes. (**B**) Insertion frequency in the Giemsa-staining-based cytobands of the *p*-arms of all nucleolar chromosomes. The panel depicts the order of the cytobands on the chromosomal arms; the central p12 bands correspond to the NORs. The stars stand for *p* value ranges of the Fisher’s exact test as follows: ****, ≤1 × 10^−4^; ***, 1 × 10^−4^–0.001; **, 0.001–0.01; n.s.—not significant. (**C**) Distribution of insertion sites in the cytobands of all acrocentric chromosomal *p*-arms. All cytobands were binned to 300 windows (data points on the *x*-axis), the normalized insertion sites are show on the *y*-axis. RPKM stands for reads per kilobase per million mapped reads. Below, the TA dinucleotide frequencies are shown in the windows specified above. The values were normalized with the window lengths, scaled to 0–1 and smoothed using the Loess method.

**Figure 4 ijms-24-14978-f004:**
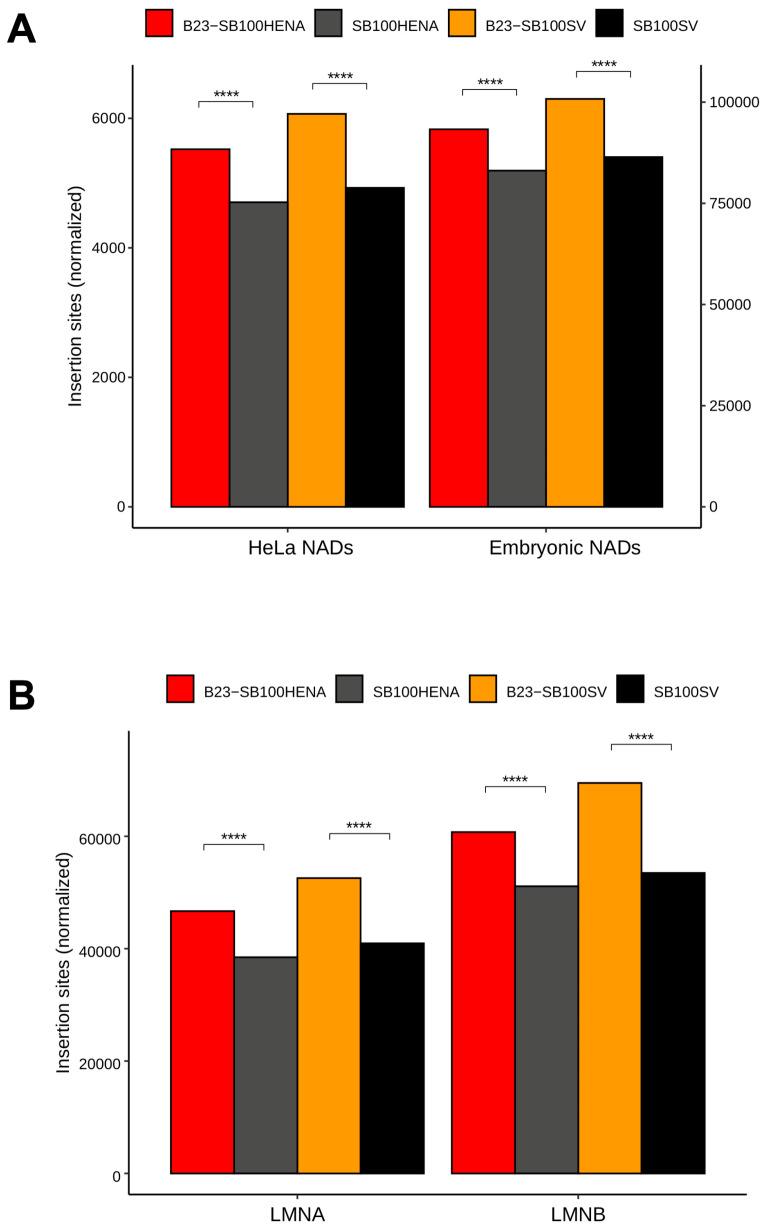
**Frequency of integrations in genome domains near the nucleolus.** (**A**) Number of insertion sites in nucleolar-associated domains (NADs) identified in HeLa cells and in human embryonic fibroblasts. (**B**) Insertion frequency in lamin-associated domains (LADs). LMNA and LMNB stand for LADs assembled around lamin A/C and lamin B1/B2, respectively. The stars stand for *p* value ranges of the Fisher’s exact test: ****, *p* ≤ 1 × 10^−4^.

**Figure 5 ijms-24-14978-f005:**
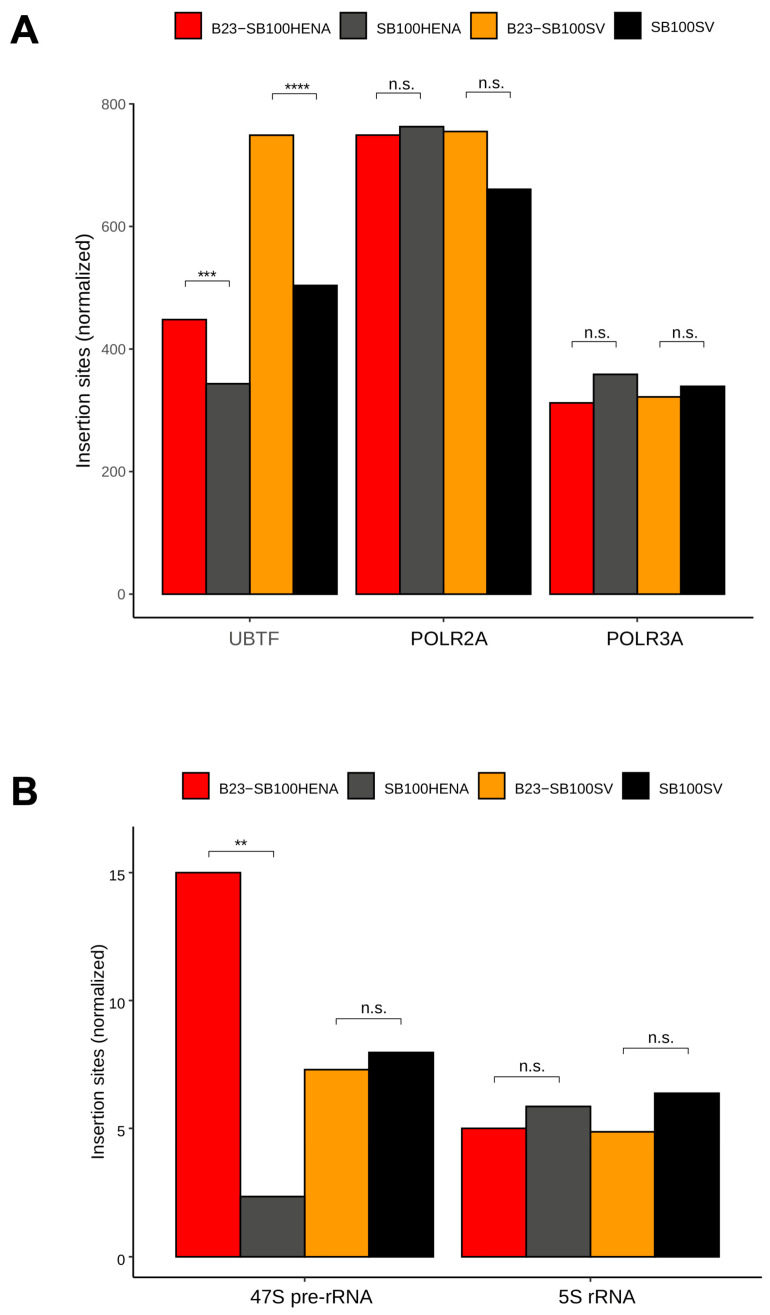
**Insertions in ribosomal DNA genes.** (**A**) Insertion numbers in transcription factor binding sites. Integrations were counted in the genomic binding regions of UBTF and the large subunits of RNA Pol II (POLR2A) and III (POLR3A). (**B**) Integrations within the RNA Pol I-transcribed 47S rRNA genes and in the RNA Pol III-driven 5S rDNA genes. The stars stand for *p* value ranges of the Fisher’s exact test as follows: ****, ≤1 × 10^−4^; ***, 1 × 10^−4^–0.001; **, 0.001–0.01; n.s.—not significant.

## Data Availability

The data presented in this study is contained within the article.

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
