# Peer review of "Sleeping Beauty Transposon Insertions into Nucleolar DNA by an Engineered Transposase Localized in the Nucleolus"

_ijms, 2023, doi:10.3390/ijms241914978_

Round 1
Reviewer 1 Report
Transposons are mobile genetic elements that can mediate the insertion of a transgenic cargo into a cell of choice. Their insertion bias has been studied in the past and is well established. Here, the authors fuse one of the best studied transposase enzymes, Sleeping Beauty, to a gene product (B23) that is preferentially localized to the nucleolus. While the study is well done and overall convincing, some minor points have to be addressed:
1. First and foremost, Figure 3 seems to be missing. Instead, it seems as if Supplementary Figure 1 had been included. As Figure 3 is of central importance, this reviewer will need to re-consider his review once the correct figure has been provided.
2. The authors mention that the tested fusions of SB with a nucleolar localization signal that did not lead to any enrichment of SB in the nucleolus, but do not show this data. At my point of view, a Supplementary Figure showing this data would be preferred
3. For the sake of transparency, it seems key to include (i) the full amino acid sequence of SB fused to B23 and (ii) the DNA sequences of the ITRs used to deliver the cargo.
4. Figure 2B shows an assay in which the authors measure transposition efficiency. Can the authors provide raw data (e.g. crystal violet-stained colonies) as a supplementary information? Also, it would be great to see how many colonies arise in the absence of the transposase enzyme.
5. Figure 2C hinges on B23 being a reliable marker for the nucleolus which this reviewer cannot judge. Can the authors provide a staining with an alternative marker? Or can the authors at least provide a reference confirming the nucleolar localization of B23?
6. Figure 2C: The authors mention that the B23-SB fusion only localizes to the nucleolus in a fraction of all cells. Can the authors provide a quantification of this as a Supplementary Information?
7. The authors use two different plasmid architectures to test their hypothesis. It would be great if the architecture could be displayed as a schematic for ease of interpretation.
8. A critical experiment is the determination of insertion sites (Figures 3-5). Can the authors please provide more explanation in the main text (results) to explain how those insertions sites have been mapped?
9. The authors mention the total number of insertion sites identified by NGS. Can this information be displayed as a graphic?
10. The authors provide various ways to assess whether nucleolar insertions have been enriched (see Figures 4 and 5) for details. While these analyses are credible, this reviewer would like to see an estimate of the proportion of insertion sites that map to nucleoli (versus all insertion sites). Can this estimate (based on Figures 4 and 5) be provided as additional Supplementary Figure?
OK
Author Response
Transposons are mobile genetic elements that can mediate the insertion of a transgenic cargo into a cell of choice. Their insertion bias has been studied in the past and is well established. Here, the authors fuse one of the best studied transposase enzymes, Sleeping Beauty, to a gene product (B23) that is preferentially localized to the nucleolus. While the study is well done and overall convincing, some minor points have to be addressed:
First and foremost, Figure 3 seems to be missing. Instead, it seems as if Supplementary Figure 1 had been included. As Figure 3 is of central importance, this reviewer will need to re-consider his review once the correct figure has been provided.
We apologize for the mixup. We have now submitted the correct figures in the correct order.
The authors mention that the tested fusions of SB with a nucleolar localization signal that did not lead to any enrichment of SB in the nucleolus, but do not show this data. At my point of view, a Supplementary Figure showing this data would be preferred.
We present these data in new Supplementary Figure 1.
For the sake of transparency, it seems key to include (i) the full amino acid sequence of SB fused to B23 and (ii) the DNA sequences of the ITRs used to deliver the cargo.
The full amino acid sequence of the B23-SB100X fusion protein is now presented in new Supplementary Figure 2A. Because the transposon sequences that we employed in our vectors are commercially available and standard vectors, we included their source in Materials and Methods, rather than spelling out the actual sequence.
Figure 2B shows an assay in which the authors measure transposition efficiency. Can the authors provide raw data (e.g. crystal violet-stained colonies) as a supplementary information? Also, it would be great to see how many colonies arise in the absence of the transposase enzyme.
We present these data in new Supplementary Figure 3. In this experiment, we concentrated our efforts on comparing the transpositional efficiencies of unfused SB100X transposase and the B23-SB100X fusion transposase. A negative control (no transposase added) is not shown in this experiment. A transposase- control is often, but not always, included in our transposition assays. Under the conditions used in our assays, approximately 10-20 antibiotic-resistant cell colonies are expected to arise. The Reviewer is kindly referred to a recent article from our group, where relative numbers of cell colonies in the absence and presence of the transposase are documented:
A single amino acid switch converts the Sleeping Beauty transposase into an efficient unidirectional excisionase with utility in stem cell reprogramming.
Kesselring L, Miskey C, Zuliani C, Querques I, Kapitonov V, Laukó A, Fehér A, Palazzo A, Diem T, Lustig J, Sebe A, Wang Y, Dinnyés A, Izsvák Z, Barabas O, Ivics Z.Nucleic Acids Res. 2020 Jan 10;48(1):316-331. doi: 10.1093/nar/gkz1119.PMID: 31777924
Figure 2C hinges on B23 being a reliable marker for the nucleolus which this reviewer cannot judge. Can the authors provide a staining with an alternative marker? Or can the authors at least provide a reference confirming the nucleolar localization of B23?
We added a reference for nucleolar localization of B23.
Figure 2C: The authors mention that the B23-SB fusion only localizes to the nucleolus in a fraction of all cells. Can the authors provide a quantification of this as a Supplementary Information?
The claim in the manuscript (“…B23-SB100X fusion localized to the nucleolus in approximately half of cells, and in the other half it was distributed similarly to unfused SB100X”) was based on our general observations during microscopy, however the number of images that we took does not allow us to deliver quantitative data, let alone establishing statistical significance.
In order to more accurately reflect that our observations deliver qualitative rather than quantitative information, we have reformulated the sentence as follows: “…B23-SB100X fusion localized to the nucleolus in only a subset of cells, while in others it was distributed similarly to unfused SB100X”).
The authors use two different plasmid architectures to test their hypothesis. It would be great if the architecture could be displayed as a schematic for ease of interpretation.
The architectures of the two different transposon constructs employed in our study are depicted in new Supplementary Figure 2B.
A critical experiment is the determination of insertion sites (Figures 3-5). Can the authors please provide more explanation in the main text (results) to explain how those insertions sites have been mapped?
We added a brief sentence in the text to provide a brief explanation of the procedure and added a reference where the procedure was used previously.
The authors mention the total number of insertion sites identified by NGS. Can this information be displayed as a graphic?
We present a new analysis displayed in Supplementary Figure S7. The figure depicts an overall genome-wide distribution of the integrations on the chromosomes.
The authors provide various ways to assess whether nucleolar insertions have been enriched (see Figures 4 and 5) for details. While these analyses are credible, this reviewer would like to see an estimate of the proportion of insertion sites that map to nucleoli (versus all insertion sites). Can this estimate (based on Figures 4 and 5) be provided as additional Supplementary Figure?
We thank the reviewer for this remark. In response, we present a new analysis displayed in Supplementary Figure S6. The figure delivers frequencies of transposon insertions within and in the vicinity of the nucleolar DNA, and displays the data as percentage of all insertions within p-arms, NADs, LADs and cumulative frequencies of these three categories.

Reviewer 2 Report
Overall, this is a well written and compelling manuscript. The idea of targeting nucleolar DNA regions is an interesting approach to reducing potential genotoxicity with an integrating vector system. The experiments are well described and interpreted. However, I do have some questions regarding the assessment of the B23-SB100X enzyme function (Figure 2). The western blot in Figure 2A shows a significant difference in protein abundance for the B23-SB100X enzyme relative to the SB100X control. This apparent difference is not mentioned nor discussed. Is this merely a difference in protein transfer due to the significant difference in the molecular weights for these enzymes? If not, then this observation needs a little more discussion and thought. Figure 2B shows that the B23-SB100X enzyme performs similarly to SB100X, at least in this assay. Is it achieving this result with less protein expressed?
I also have a general concern regarding the approach used to compare B23-SB100X with SB100X in Figure 2B. How confident are the authors that the expression of the transgene from the nucleolar genomic regions is equivalent to those integrated into non-nucleolar regions? The conclusion assumes that they are equivalent, but the number of colonies could be under or over estimated if the transgene is expressed differently from the nucleolar regions. If this is the case, the B23-SB100X enzyme has significantly different efficiency than SB100X, and that is not what the authors have concluded. One way to potentially address this is to evaluate the data in Figures 4 and 5. The data are currently shows as normalized insertions. Can the authors evaluate these data to meaningfully assess the total number of insertions per cell that each enzyme is producing? Or can the authors compare the rate of transposon excision induced by each enzyme to make a more direct comparison between the two?
Author Response
Overall, this is a well written and compelling manuscript. The idea of targeting nucleolar DNA regions is an interesting approach to reducing potential genotoxicity with an integrating vector system. The experiments are well described and interpreted. However, I do have some questions regarding the assessment of the B23-SB100X enzyme function (Figure 2). The western blot in Figure 2A shows a significant difference in protein abundance for the B23-SB100X enzyme relative to the SB100X control. This apparent difference is not mentioned nor discussed. Is this merely a difference in protein transfer due to the significant difference in the molecular weights for these enzymes? If not, then this observation needs a little more discussion and thought. Figure 2B shows that the B23-SB100X enzyme performs similarly to SB100X, at least in this assay. Is it achieving this result with less protein expressed?
We thank the reviewer for bringing up this point. We think the difference in signal intensities could be due to several factors (or a combination of them) that do not necessarily reflect a generally reduced expression of B23-SB100X when compared to SB100X.
- a) For the WB, the cells were transfected with a fixed mass of DNA, rather than equimolar amounts. Due to the relatively larger size of the B23-SB100X expression plasmid (as compared to the SB100X plasmid), this means that a lower number of plasmid molecules were transfected, which might contribute to lower expression.
- b) In addition to point a), the generally lower transfection efficiency of larger plasmids likely further reduces the delivery of that construct.
- c) Finally, as mentioned by the Reviewer, the lower transfer efficiency of larger proteins in the WB itself might contribute to the difference in signal intensities.
Thus, given the setup of the experiment, we cannot really make a claim about whether one construct is expressed more efficiently than the other. However, we consider it likely that the observed difference in signal intensity is due to these technical aspects.
We would like to stress that the Western blot presented in Fig. 2A was meant to deliver qualitative information (fusion protein is expressed and has the expected size) rather than quantitative information (relative levels of proteins). The Reviewer suggests a reasonable conclusion that lower amounts of the B23-SB100X fusion protein can catalyze almost equal levels of transposition as unfused SB100X transposase. However, we should keep in mind that transposition efficiencies cannot be solely calculated based on colony numbers. Transposition efficiencies could be calculated from colony numbers and average integrated transposon copy numbers per colony (or per cell in which transposition occurred). Because copy numbers were not measured in our experiments, we do not further comment on the relative efficiencies of transposition by B23-SB100X versus SB100X. Again, the intention of the experiment was to demonstrate that the B23-SB100X fusion protein is transpositionally active (which is a prerequisite for all of the subsequent experiments) rather than claiming its relative efficiency.
I also have a general concern regarding the approach used to compare B23-SB100X with SB100X in Figure 2B. How confident are the authors that the expression of the transgene from the nucleolar genomic regions is equivalent to those integrated into non-nucleolar regions? The conclusion assumes that they are equivalent, but the number of colonies could be under or over estimated if the transgene is expressed differently from the nucleolar regions. If this is the case, the B23-SB100X enzyme has significantly different efficiency than SB100X, and that is not what the authors have concluded. One way to potentially address this is to evaluate the data in Figures 4 and 5. The data are currently shows as normalized insertions. Can the authors evaluate these data to meaningfully assess the total number of insertions per cell that each enzyme is producing? Or can the authors compare the rate of transposon excision induced by each enzyme to make a more direct comparison between the two?
We thank the reviewer for this remark, which points to a similar direction of a point Reviewer 1 has raised. We present a new analysis displayed in Supplementary Figure S6. The figure delivers frequencies of transposon insertions within and in the vicinity of the nucleolar DNA, and displays the data as percentage of all insertions within p-arms, NADs, LADs and cumulative frequencies of these three categories. The figure demonstrates that it is only a small fraction of the total insertions, which mapped near to the arrays of rRNA genes, and that the majority of the insertion events took place in the nucleolus-associated DNA. We also discuss in the main text of the manuscript potential reasons for the transposon to actually disfavor integration in the NORs. Our findings are supported by the finding that enrichment in NADs and LADs were observed for both transposons (containing either Pol I or Pol II promoters) in the presence of the B23-SB100X fusion transposase, suggesting that expression of transgenes situated in these genomic regions by these promoters is possible. We conclude that the numbers of insertions that we received were not significantly skewed by the different promoters employed in the transposon vectors.

Reviewer 3 Report
The manuscript describes an interesting work aimed at the use of the rDNA loci as safe harbours for foreign DNA integration.
I cannot submit a full report at this stage, since I can only partially evaluate the manuscript. Indeed, figure 3 is missing. It has been actually replaced with supplementary figure 1. Since figure 3 appears to be essential for understanding and evaluating the results described in the rest of the manuscript, I will submit my report after receiving a correct version of the manuscript.
I look forward to receive your full submission.
Although it seems to be fine, I will express my opinion in my final report
Author Response
We apologize for the mistake. I include proper Fig. 3 now.

Round 2
Reviewer 3 Report
The authors have explored the possibility of targeting SB integration to the rDNA loci through fusion-mediated tethering to the SB transposase and the use of a POL1 promoter to drive the reporter expression. While they observed a significant increase in the number of insertions, they also found a positive correlation with insertions occurring in NADs and in chromatin associated with LMNA and LMNB. This is an important study aimed at identifying new safe harbors for gene therapy.
The manuscript is very well written and I have few minor comments. Line numbering in my report refer to the revised PDF file.
L111 Pol I-driven genes -> RNApolI-driven genes
L112 different location -> different gene (or loci)
L199 antibiotic-tagged transposon is not the correct way to refer to the selection marker carrying transposon
Figure 3. does the second Y axis also refer to the normalized insertion sites? Please clarify
Results: Since the number of insertion/cell is an important parameter to take into account to determine proper expression levels, I wonder if it would be possible to establish the number of insertion per cell from the experimental results. Given the redundancy of the rDNA it would be expected that more than one insertion occurs per cell.
Discussion: Silencing is an important phenomenon occurring at the rDNA loci.
While the authors claim that targeting rDNA would haves several advantages including preventing transgene silencing, the authors should discuss the possible role of nucleolar dominance phenomenon and how it can impact both the efficiency of transposition and trans gene expression in the NOR
Figures: At least in the PDF file, the resolution of all figures is very low and must be improved
Author Response
The manuscript is very well written and I have few minor comments. Line numbering in my report refer to the revised PDF file.
We thank the reviewer for her/his overall positive remarks on our manuscript.
L111 Pol I-driven genes -> RNApolI-driven genes
We made the correction here and everywhere else in the manuscript.
L112 different location -> different gene (or loci)
We made the correction.
L199 antibiotic-tagged transposon is not the correct way to refer to the selection marker carrying transposon
Thank you, correctly noted. We made the necessary correction.
Figure 3. does the second Y axis also refer to the normalized insertion sites? Please clarify
We refined the figure legend as follows: “Insertion numbers are normalized with the total number of insertions per condition, shown on both axes. The second y-axis applies to the insertion sites on the non - acrocentric chromosomes.”
Results: Since the number of insertion/cell is an important parameter to take into account to determine proper expression levels, I wonder if it would be possible to establish the number of insertion per cell from the experimental results. Given the redundancy of the rDNA it would be expected that more than one insertion occurs per cell.
We have no means to determine the actual numbers of insertions in these particular experiments, but we refer to an earlier study by us where we determined SB transposon copy numbers by clonal analysis in HeLa cells, transfected by similar amounts of transposon and transposase plasmids. We touch upon the fact that we do expect multiple transposon integrations per cells in lines 463-466 of the revised manuscript.
Discussion: Silencing is an important phenomenon occurring at the rDNA loci.
While the authors claim that targeting rDNA would haves several advantages including preventing transgene silencing, the authors should discuss the possible role of nucleolar dominance phenomenon and how it can impact both the efficiency of transposition and trans gene expression in the NOR
We thank the reviewers for bringing up this consideration. We inserted a discussion, along with new references, of this phenomenon in lines 452-462 of the revised manuscript.
